# Epidemiological Studies of Pan-Azole Resistant *Aspergillus fumigatus* Populations Sampled during Tulip Cultivation Show Clonal Expansion with Acquisition of Multi-Fungicide Resistance as Potential Driver

**DOI:** 10.3390/microorganisms9112379

**Published:** 2021-11-18

**Authors:** Bart A. Fraaije, Sarah L. Atkins, Ricardo F. Santos, Steven J. Hanley, Jonathan S. West, John A. Lucas

**Affiliations:** 1NIAB, Cambridge CB3 0LE, UK; sarah.atkins@niab.com; 2Rothamsted Research, Harpenden AL5 2Q, UK; steve.hanley@rothamsted.ac.uk (S.J.H.); jon.west@rothamsted.ac.uk (J.S.W.); john.lucas@rothamsted.ac.uk (J.A.L.); 3Luiz de Queiroz College of Agriculture, University of São Paulo, Piracicaba 13418-900, SP, Brazil; ricardofelicianodossantos@gmail.com

**Keywords:** fungicide resistance, azole fungicides, fungicide target proteins, CYP51A, aspergillosis, *Aspergillus fumigatus*, clonal lineages

## Abstract

Pan-azole resistant isolates are found in clinical and environmental *Aspergillus fumigatus* (*Af*) populations. Azole resistance can evolve in both settings, with *Af* directly targeted by antifungals in patients and, in the environment, *Af* unintendedly exposed to fungicides used for material preservation and plant disease control. Resistance to non-azole fungicides, including methyl benzimidazole carbamates (MBCs), quinone outside inhibitors (QoIs) and succinate dehydrogenase inhibitors (SDHIs), has recently been reported. These fungicide groups are not used in medicine but can play an important role in the further spread of pan-azole resistant genotypes. We investigated the multi-fungicide resistance status and the genetic diversity of *Af* populations sampled from tulip field soils, tulip peel waste and flower compost heaps using fungicide sensitivity testing and a range of genotyping tools, including STR*Af* typing and sequencing of fungicide resistant alleles. Two major clones were present in the tulip bulb population. Comparisons with clinical isolates and literature data revealed that several common clonal lineages of TR_34_/L98H and TR_46_/Y121F/T289A strains that have expanded successfully in the environment have also acquired resistance to MBC, QoI and/or SDHI fungicides. Strains carrying multiple fungicide resistant alleles have a competitive advantage in environments where residues of multiple fungicides belonging to different modes of action are present.

## 1. Introduction

Airborne *Aspergillus fumigatus* (*Af*) spores can, after inhalation, cause disease in animals and humans ranging from allergic conditions and chronic lung infection to acute invasive aspergillosis (IA). Patients with a weakened immune system are most at risk for IA, and successful therapy depends on early diagnosis and the effective use of antifungals. Due to their efficacy and low toxicity, orally administered azoles have been the drugs of choice, followed by intravenous applications of amphotericin B and echinocandins. However, azole resistance has evolved in *Af* and is becoming more common in both the clinical setting and the wider environment since the late 1990s [1,2,3]. Further spread of resistance will have a significant clinical impact as higher mortality rates have already been recorded for patients with voriconazole-resistant IA in comparison with voriconazole-susceptible infection [4].

Azoles inhibit the enzyme sterol 14α-demethylase (CYP51), a key step in the synthesis of sterols essential for the integrity of cell membranes. Although different azole resistance mechanisms are known [5,6,7], azole resistance in clinical and environmental *Af* isolates is mainly associated with alterations in the regulatory and/or coding region of the CYP51A paralogue [8,9]. Azole resistance developed during azole therapy is often associated with single mutations, including G54A/E/RV/W, P216L, M220I/K/L/R/T/V and G448S, conferring different levels of resistance to different azoles. Most highly multi-azole resistant strains found both in clinical settings and the environment belonged to three unique, more complex genotypes—TR_34_/L98H, TR_34_/L98H/S297T/F495I and TR_46_/Y121F/T289A. These CYP51A variants are based on a combination of promoter tandem repeat (TR) insert of 34 or 46 bp with mutations resulting in amino acid substitutions L98H, Y121F, S297T, T289A and F495I. Isolates carrying TR_34_/L98H have been found in Europe since 1998 [10]. TR_34_/L98H/S297T/F495I, first detected in the Netherlands in 1998 [2], is often found in Asia [11,12], whereas the first TR_46_/Y121F/T289A isolate was reported from North America in 2008 [13]. More complex CYP51A variants have recently emerged, including TR_34_/L98H/T289A/I364V/G448S, TR_46_/Y121F/T289A/S363P/I364V/G448S and different TR_46_ variants (TR_46_^2^, TR_46_^3^ and TR_46_^4^) in combination with Y121F, M172V, T289A and G448S [14,15,16,17]. 

Recent studies have shown that several TR_34_- and TR_46_-based CYP51A variants have evolved resistance to non-azole fungicides belonging to different modes of action that are not used in medicine [17,18,19]. These fungicides, methyl benzimidazole carbamates (MBC), quinone outside inhibitors (QoI) and succinate dehydrogenase inhibitors (SDHI), targeting β-tubulin (cytoskeleton), cytochrome *b* (respiration) and succinate dehydrogenase (respiration), respectively, are commonly used to control plant diseases and resistance development in *Af* is an unintended side effect as agronomically used azole fungicides are not directly targeted against *Af*. Although TR_34_/TR_46_-based CYP51A have been associated with the environmental route of resistance development rather than the patient route, the origin of these strains without additional CYP51A mutations remains unclear and an initial selection in the clinical setting with subsequent spread through aerosols into the wider environment cannot be excluded [20,21]. However, fungicide resistance can evolve in the environment and measures should be taken to prevent further spread to retain the effectiveness of the azole class for both environmental and medical applications [22]. 

Research is ongoing to determine the hotspots for resistance development and spread. Tulip bulbs have previously been identified as a vehicle for international spread of azole resistant *Af* isolates [23,24]. Decaying flower bulb waste from farms, industrial wood-chip waste, and industrial green-waste storage, able to support the growth and reproduction of *Af* in the presence of fungicides, have already been identified as hotspots [25]. A better understanding on the emergence and spatiotemporal spread of different fungicide resistant alleles can improve our understanding on hotspots for fungicide resistance development and aid development of measures to reduce exposure risk. 

The aim of this study was to investigate the resistance status of *Af* populations sampled from tulip field soils, tulip bulbs, tulip peel waste piles and flower compost heaps to azole, MBC, QoI and SDHI fungicides. A panel of clinical isolates and a set of environmental isolates, that were characterized in a previous study [17], were also included to compare the fungicide sensitivity phenotypes and genotypes. Sequencing of genes encoding fungicide target proteins, cell surface protein (CSP) and mating type, as well as microsatellite typing based on short tandem repeats (STR*Af*) was carried out for all strains to identify genetical relatedness and to check for clonal expansion [26,27]. The spread of clonal lineages in both the environment and the clinical setting is further discussed using data that can be found in the literature or the available AfumID database [28].

## 2. Materials and Methods

### 2.1. Sampling and Strain Isolation

In total, 32 samples, ranging from tulip field soils (soil not in close contact with bulbs was sampled), flower bulbs, tulip peel waste heaps (decaying material) and flower waste compost sites were studied (Table 1). Soils from six tulip fields at five locations were sampled to a depth of 8 cm at three sampling points, separated five meters apart during summer in 2016. 

Materials from tulip peel waste heaps and compost sites were sampled and sent in by commercial tulip growers during August 2018. For both soil, bulb peel waste heap and compost samples, 2 g aliquots of material were added to 8 mL of phosphate buffered saline amended with 0.1% (*v*/*v*) Tween 20 (PBST) and *Af* strains isolated as described in our previous study [17]. Tulip and daffodil bulbs were purchased directly from garden centers or ordered on-line from nurseries in the autumn during 2016–2018. The outer skin of ten bulbs per sample was peeled off, pooled, and further processed using 25 mL of PBST. To grow and isolate *Af* colonies aliquots of the buffer extracts were plated out on untreated, or azole fungicide (5 µg/mL tebuconazole) amended Sabouraud dextrose (SD) agar (Oxoid Ltd., Basingstoke, UK) containing penicillin (100 U/mL) and streptomycin (100 μg/mL) and incubated for 2 days at 48 °C.

### 2.2. Panel of Clinical Isolates 

A study of a panel of 20 *Af* clinical strains, which included 18 azole-insensitive and two reference strains (AF65 (National Collection of Pathogenic Fungi (NCPF) 7097)) and AF293 (NCPF 77367)), all isolated before 2018, was also caried out to allow comparisons of the fungicide sensitivity pheno- and genotypes found in the clinical setting and in the environment. The isolates originated from Belgium (CYP_15_2, 15_7, 15_38, 15_46, 15_63 and 15_80), Germany (Asp 164, 168, 251 and 267), Japan (OKH50 (2016)), Taiwan (D007), the Netherlands (ARAF013, ARAF017, V093-26 (2010) and V094-54 (2009) and the UK (AF65 (1997), AF293 (1993), CXH_06 and 07). 

### 2.3. Fungicide Sensitivity Testing

After subculturing single colonies in tissue culture flasks with 12 mL SD agar for seven days at 37 °C, spores were harvested through shaking with 5 mm glass beads after addition of 3 mL of saline. Spore suspensions containing approximately 10^6^ spores/mL in sterile distilled water were used in the microprocessor controlled Autoplate Spiral Plating System AP5000 (Advanced Instruments, Norwood, MA, USA) according to the manufacturer’s instructions as described previously [17]. Depending on the molecular weight of the compounds, an up to a 200-fold fungicide dilution gradient on SD agar was achieved using 15 cm plates. The concentration ranges (µg/mL) for the different fungicides were: boscalid (0.1–18.469), carbendazim (0.1–11.464), imazalil (0.25–43.153), itraconazole (0.025–6.113 or 0.1–22.324), posaconazole (0.005–1.0) (pyraclostrobin (0.1–20.120), tebuconazole (0.1–17.349), terbinafine (0.01–1.7) and voriconazole (0.1–19.120). The different concentration ranges were chosen to distinguish sensitive wild-type (wt) isolates without known resistance mechanisms from those of insensitive isolates harboring resistance mechanism(s) in a single assay. Isolates were streaked on the spiral SD agar plates (eight per plate) from the outside to the center using cotton swaps and incubated at 37 °C in the dark. After 24 h incubation, the fungal growth of each isolate on the spiral plate was visually assessed and MIC values determined using the Spiral Gradient Endpoint (SGE) software.

### 2.4. DNA Extractions

After harvesting spores from one-week cultures in tissue culture flasks, 1.5 mL of spore suspensions was transferred into a 2 mL tube and centrifuged for 2 min at 13,200 rpm. After removing the supernatant, DNA was extracted according to the MasterPure Yeast DNA Purification kit (Lucigen Corporation, Madison, WI, USA) with the inclusion of an extra bead-beating step, which involved the addition of glass beads (0.425–0.600 mm) and the use of the Genie 2 Vortex (Scientific Industries, Bohemia, NY, USA) at full power for 2 min. This step was carried out after the lysis step just before adding the MPC Protein Precipitation Reagent. 

### 2.5. PCR Amplification and Sequencing of Fungicide Resistant Alleles

Whole or partial fungicide target genes (β-tubulin, cytochrome *b* and succinate hydrogenase subunit B), covering all regions where mutations affecting inhibitor binding have been reported for fungi, were amplified with PCR, sequenced, and analyzed using Geneious software version 10.0 (Biomatters, Auckland, New Zealand) according to the previously established protocols [17]. 

### 2.6. Cell Surface Protein Typing

The cell surface protein (CSP) (XM_749624.1) encoding gene of *Af* was also partially sequenced from selected strains [29]. CSP typing of strains, including CSP t02* or t02B [17], was carried out according to the established nomenclature [26], which is based on the tandem repeat region, in which up to ten different 12-bp repeat sequences have been found in different copy numbers, and the flanking regions. After manual alignment of sequences, a phylogenetic tree for the different CSP sequences encountered in this study was constructed using the Geneious Tree Builder software (Biomatters, Auckland, New Zealand) based on the Tamura-Nei distance model and the Neighbor-Joining method.

### 2.7. Microsatellite Typing Based on Short Tandem Repeats in A. fumigatus (STRAf)

In total, 128 *Af* isolates with different levels of azole sensitivity from this study and our previous studies on environmental and reference isolates were further characterized using microsatellite genotyping based on a panel of nine short tandem repeat markers (STR*Af* 2A, 2B, 2C, 3A, 3B, 3C, 4A, 4B and 4C) according to the method previously described and validated [27,30]. GenAlEx v.6.503 software was used to identify distinct multi-locus genotypes (MLGs), where a single difference in an allele size was considered enough to discriminate a unique MLG [31]. To visualize the phylogenetic relationships among MLGs, a minimum spanning network (MSN) was generated using the R package poppr v.2.8.3 [32], which is based on Bruvo’s distance [33]. 

## 3. Results

### 3.1. Isolation and Fungicide Sensitivity Testing of A. fumigatus Isolates from Tulip Field Soils

In total, 180 isolates were isolated from soils sampled from six fields at five different locations (Table 1). Figure 1 and Figure 2 show the sensitivity profiles for the overall population to eight different fungicides.

The frequency of pan-azole resistant isolates within this population varied between 0% and 16.7% per field (*n* = 30) with an average overall frequency of 5.6% (10 out of 180) (Table 1). Amongst the 10 pan-azole resistant isolates, three isolates from these (STNL5-B7, STNL5-C1 and STNL5-C8), all isolated from tulip field 5 (Table 1), carried TR_46_/Y121F/T289A and showed high levels of insensitivity to all azoles, itraconazole (>22.324 µg/mL), voriconazole (>19.120 µg/mL), posaconazole (>0.200 µg/mL), imazalil (>15.0 µg/mL) and tebuconazole (>17.349 µg/mL), carbendazim (>11.464 µg/mL) and pyraclostrobin (>20.120 µg/mL). Six out of seven isolates, showing lower levels of insensitivity to most of the azoles tested, carried TR_34_/L98H. Four of these (STNL2-B8, STNL5-B6, STNL6-A3 and STNL6-B2) were sensitive to both carbendazim and pyraclostrobin (MIC values below 3.0 µg/mL), while STNL3-C8 was highly insensitive to pyraclostrobin but not carbendazim, and STNL6-B1 highly insensitive to both carbendazim and pyraclostrobin. The remaining isolate, STNL5-C5, with low levels of insensitivity to multiple azoles carried TR_34_/L98H/S297T/F495I and was also insensitive to carbendazim.

Only one azole sensitive strain, STNL1-A8 with wild-type CYP51A, was resistant to carbendazim. Several strains showing low levels of insensitivity to tebuconazole, in particular, carried F46Y/M172V/E427K. Amongst these strains, STNL2-C9 showed a raised MIC value of 6.017 µg/mL to pyraclostrobin. As expected, only small differences in sensitivity to terbinafine, a squalene epoxidase inhibitor, were measured for all isolates and this was not linked to a CYP51A variant. Of the medical azoles tested, posaconazole was most effective with the majority of isolates displaying MICs between 0.007 and 0.046 µg/mL, followed by voriconazole (most strains in MIC range 0.100 to 0.657 µg/mL), imazalil (most strains in MIC range 0.250 to 1.643 µg/mL), tebuconazole (most strains in MIC range 0.110 to 1.710 µg/mL) and itraconazole (most strains in MIC range 0.403 to 1.774 µg/mL), respectively (Figure 1). A wide range of sensitivities were measured for itraconazole with a large fraction of isolates (*n* = 33), the majority carrying wild-type CYP51A alleles, having MIC values exceeding 10.000 µg/mL. 

### 3.2. Isolation and Fungicide Sensitivity Testing of A. fumigatus Isolates from Flower Bulbs, Tulip Peel Waste Heaps and Compost

In total, 200 strains were isolated from tulip bulbs (*n* = 128), daffodil bulbs (*n* = 30) and tulip peel waste (*n* = 42) on SD agar without addition of tebuconazole and 19 strains from compost (*n* = 19) on tebuconazole-amended SD agar (Table 1). Eight additional strains were also isolated from three different tulip peel waste samples using tebuconazole-amended SD agar. The results of the fungicide sensitivity tests for the four largest populations together with the tulip soil population are displayed in Figure 2, while test results for a selection of individual strains are shown in Table 2. 

As expected, 19 compost isolates (Table 1), all isolated from tebuconazole-amended SD agar, showed high levels of insensitivity to all azoles tested, with 18, 17 and 14 of these also having MIC values exceeding 17.349, 19.120 and 43.153 µg/mL, for tebuconazole, voriconazole and imazalil, respectively (Figure 2). All 19 isolates were also highly insensitive to both carbendazim and pyraclostrobin having MIC values greater than 11.464 and 20.120 µg/mL, respectively. 

Tulip peel waste heaps also contained high frequencies of multi-fungicide insensitive isolates, with 23 out of 42 strains tested insensitive to multiple azoles and pyraclostrobin, respectively (Figure 2). Of these, 22 were also highly insensitive to carbendazim. Six out of 23 were highly insensitive to voriconazole, tebuconazole and imazalil showing MICs greater than 10 ppm (including TP UT4A-1 and TP UT4B-2), while the remaining isolates (including TP UT1B-1, TP UT1A-2 and TP UT5C-5), showed raised levels of insensitivity to one or more azoles. All 19 isolates that were sensitive to pyraclostrobin were also sensitive to all azoles tested with MICs lower than 1.0, 2.0 and 4.0 µg/mL for voriconazole, imazalil and tebuconazole, respectively. All eight strains isolated from tebuconazole-amended SD agar were insensitive to both carbendazim and pyraclostrobin and showed high levels of insensitivity to multiple azoles (including TP TEB5C-5).

The frequency of multi-fungicide resistant isolates was much lower for the sampled population from tulip bulbs (Figure 2). Only eight out of 128 isolates were highly insensitive to carbendazim, of which seven (T3-5, T4-9, T4-10, T5-1, T5-2, T5-5 and T7-9) were also highly insensitive to voriconazole, tebuconazole and imazalil, showing MICs greater than 10 µg/mL. Reduced sensitivity to two or more azoles (MICs exceeding 1.0, 2.0 and 4.0 µg/mL for voriconazole, imazalil and tebuconazole, respectively) were measured for the other carbendazim insensitive isolate (T11-8) and 17 additional isolates (including T2-1, T3-6 and T6-3). None of the strains isolated from daffodil bulbs showed any levels of insensitivity to carbendazim, voriconazole and tebuconazole while only one isolate tested positive for imazalil, with a MIC of 2.051 µg/mL just above the discriminatory dose of 2.0 µg/mL. 

### 3.3. Azole Resistance Phenotype-to-Genotype Relationship, Cell Surface Protein and Mating Typing of A. fumigatus Isolates from the Environment with a Focus on Tulip Cultivation

A selection of 30 environmental *Af* isolates with different levels of insensitivity to azoles, carbendazim and/or pyraclostrobin were further tested for sensitivity to boscalid and were further characterized by determination of CSP, mating type and fungicide resistant alleles (Table 2).

Isolates carrying TR_46_/Y121F/T289A, frequently detected in this study, and TR_34_/L98H/T289A/I364V/G448S, only detected in two strains isolated from tulip peel waste (TP TEB5C-5) and tulip compost (TC TEB6B-1) on tebuconazole-amended SD agar were highly insensitive to all azoles tested, with high MIC values greater than or equal to 19.120, 17.349 and 16.901 µg/mL measured for voriconazole, tebuconazole and imazalil, respectively (Table 2). Isolates carrying TR_34_/L98H showed lower levels of azole insensitivity to voriconazole and imazalil, but six out of 13 were also highly insensitive to tebuconazole (MIC values > 17.349 µg/mL). STNL5-C5, carrying TR_34_/L98H/S297T/F495I showed a low level of insensitivity to voriconazole and tebuconazole, but its insensitivity to imazalil (MIC value of 8.367 µg/mL) was high in comparison to most TR_34_/L98H isolates. Isolate STNL2-C9, carrying F46Y/M172V/E427K, showed only slightly raised MIC values for voriconazole, imazalil and tebuconazole in comparison with wild-type CYP51A strains. 

All TR_46_/Y121F/T289A and TR_34_/L98H/T289A/I364V/G448S isolates, as well as four out of 13 TR_34_/L98H isolates tested, were also insensitive to both carbendazim (MIC values > 11.464 µg/mL) and pyraclostrobin (MIC values > 20.120 µg/mL). Insensitivity to carbendazim but not to pyraclostrobin was also measured for STNL1-A8 and STNL5-C5, carrying wild-type CYP51A and TR_34_/L98H/S297T/F495I, respectively. Sensitivity to carbendazim and a moderate level of pyraclostrobin insensitivity (MIC value of 6.017 µg/mL) was measured for isolate STNL2-C9, carrying F46Y/M172V/E427K. High levels of insensitivity to pyraclostrobin but not to carbendazim were measured for TR_34_/L98H isolates STNL3-C8 and TP UT5C-5. Resistance to boscalid (MIC values > 18.469 µg/mL) was found in two isolates with high insensitivity to the different azoles, carbendazim and pyraclostrobin, being STNL6-B1 and TC TEB6B-1 carrying TR_34_/L98H and TR_34_/L98H/T289A/I364V/G448S, respectively.

β-tubulin sequence analysis revealed that insensitivity to carbendazim was conferred in all environmental isolates by the amino acid substitution F200Y (codon change TTC to TAC). Isolate STNL5-C5, carrying TR_34_/L98H/S297T/F495I, showed a double nucleotide change resulting in F200Y (TTC to TAT). High levels of pyraclostrobin insensitivity (MIC values > 20.120 µg/mL) were associated with the cytochrome *b* alteration G143A (GGT to GCT), whereas the lower level of insensitivity to pyraclostrobin (MIC value of 6.017 µg/mL) in isolate STNL2-C9 was linked to cytochrome *b* F129L (TTC to TTA). Insensitivity to boscalid was conferred by SdhB alteration H270Y (CAC to TAC) in STNL6-B1 and TC TEB6B-1.

Multiple CSP and mating types were found in both TR_46_/Y121F/T289A (CSP t01, t02 and t06A) and TR_34_/L98H isolates (CSP t01, t02, t04B and t11). CSP t02 and t02B were detected in STNL5-C5 (TR_34_/L98H/S297T/F495I) and STNL2-C9 (F46Y/M172V/E427K), respectively, whereas both TR_34_/L98H/T289A/I364V/G448S isolates carried CSP t02 and MAT1-2.

### 3.4. Azole Resistance Phenotype-to-Genotype Relationship, Cell Surface Protein and Mating Typing of A. fumigatus Isolates from the Clinical Setting

Results for the panel of 20 clinical Af isolates, including AF65 and AF293 as reference isolates, are presented in Table 3. The order of the voriconazole sensitivity mirrored the results for the environmental isoles with the reference isolates AF65 (wild-type CYP51A) and AF293 (F46Y/M172V/N284T/D255E/E427K) most sensitive, followed by isolates carrying TR_34_/L98H/S297T/F495I, TR_34_/L98H and TR_46_/Y121F/T289A, respectively. With regard to imazalil, four isolates with TR_34_/L98H/S297T/F495I showed a higher level of insensitivity, approximately three-fold, than most TR_34_/L98H isolates. In comparison with the MIC value of the wild-type isolate AF65, the average MIC value of the TR_34_/L98H/S297T/F495I isolates was approximately 15-fold higher (Resistance Factor (RF) of 15). TR_46_/Y121F/T289A were most insensitive, with RF > 25. TR_34_/L98H and TR_34_/L98H/S297T/F495I isolates showed different levels of insensitivity to tebuconazole (MIC values ranging from 3.341 to >17.349 µg/mL) with several strains also highly insensitive to carbendazim (MIC values > 11.464 µg/mL). Two TR_34_/L98H isolates, ARAF017 and CYP_15_46, were insensitive to fungicides belonging to all four different modes of action (azoles, MBC and QoI and SDHI fungicides). All six TR_46_/Y121F/T289A isolates showed high levels of insensitivity to tebuconazole and were also higly insensitive to both carbendazim (MIC values > 11.464 µg/mL) and pyraclostrobin (MIC values > 20.120 µg/mL). One isolate, CYP_15_46, was also insensitive to boscalid (MIC value > 18.469 µg/mL).

β-tubulin sequence analysis revealed that insensitivity to carbendazim was conferred in all but one isolate by the amino acid substitution F200Y (codon change TTC to TAC). Carbendazim insensitivity in isolate CYP_15_46 was conferred by the beta-tubulin alteration E198A (GAG to GCG). Like the enviromental strains, mutations resulting in cytochrome *b* G143A (GGT to GCT) and SdhB H270Y (CAC to TAC) were found in all isolates insensitive to carbendazim and pyraclostrobin, respectively. 

Multiple CSP and mating types were found in TR_46_/Y121F/T289A (CSP t01, t02 and t09), TR_34_/L98H isolates (CSP t02, t04A, t04B and t11) and TR_34_/L98H/S297T/F495I (CSP t01, t02, t04A and t11). 

### 3.5. Genetic Diversity of Azole-Resistant Aspergillus fumigatus Isolated from the Clinical Setting and the Wider Environment with a Focus on Tulip Cultivation

In total, 128 Af isolates were analysed using STRAf typing. These originated from air (*n* = 5) and arable soils (*n* = 43) as part of our previous studies, and from tulip field soils (*n* = 13), tulip bulbs (*n* = 29), tulip peel waste (*n* = 13), tulip compost (*n* = 1) and patients (*n* = 18) as described here. Six reference strains that were used in both studies were also included. Most isolates were also further characterised with CSP and mating gene typing as well as CYP51A sequence analysis (see Appendix A for details). The STRAf profile for tulip compost isolate TC TEB6B-1 was inconclusive, showing two different product sizes for six out of the nine loci, indicative of a mixed culture, and was excluded for further analysis. Among 127 isolates, 99 distinct unique multilocus genotypes (MLGs) were identified, of which 88 were represented by single isolates (Figure 3). 

Eleven MLGs were detected in multiple strains (clusters A to K), which can be regarded as clonal lineages. Cluster I was the largest cluster with ten isolates, all carrying TR_46_/Y121F/T289A, CSP t01 and MAT1-1, and insensitive to both carbendazim and pyraclostrobin (Table 2). This cluster included seven strains isolated from tulip bulbs originating from the Netherlands and the UK (T3-5, T4-9, T4-10, T5-1, T5-2, T5-5 and T7-9), two strains from tulip waste heaps (including TP UT4A-1) and reference strain, TR46-NL, isolated from the environment in the Netherlands and tested previously [17]. Cluster C consisted of nine isolates carrying TR_34_/L98H, CSP t11 and MAT1-1, with no insensitivity observed for carbendazim and pyraclostrobin. Eight isolates originated from multiple tulip bulb samples obtained in both the Netherlands and the UK (T1-2, T2-1, T3-3, T3-6, T4-7, T5-3, T6-3 and T10-5) and a soil isolate, SS8-7A, originating from a sugar beet field in Belgium in our previous study [17]. Cluster D contained two reference strains (TR34-NL and 08-19-02-10, isolated in the Netherlands in 2008) and arable soil isolate RN8-18, carrying TR_34_/L98H, CSP t04B and MAT1-2, all showing sensitivity to carbendazim and pyraclostrobin in our previous study [17]. Cluster E is formed by two clinical strains, CYP_15_7 and V093-54, from Belgium and the Netherlands, respectively, and STNL5-C8, a Dutch tulip field soil isolate, all carrying TR_46_/Y121F/T289A, CSP t01 and MAT1-2, all showing insensitivity to both carbendazim and pyraclostrobin. The remaining clusters contained two isolates. Cluster F contained strain TP UT5C-5, isolated from a tulip peel waste heap sample, and clinical strain OKH50 isolated in Japan in 2016 [34], both carrying TR_34_/L98H, CSP t02 and MAT1-2. Both isolates were sensitive to carbendazim, but only TP UT5C-5 was insensitive to pyraclostrobin. Two tulip waste heap isolates, including TP UT1B-1, formed cluster H, both carrying TR_34_/L98H, CSP t11 and MAT1-1, and insensitive to carbendazim and pyraclostrobin. Clusters J and K are formed by F46Y/M172V/E427K isolates with low levels of insensitivity to azoles detected previously (ref) and carried combinations of CSP t13 with MAT1-2 and CSP t02B with MAT1-1, respectively. All F46Y/M172V/E427K isolates grouped together in the upper right branch positioned above cluster J in Figure 3. With the exception of isolate STNL2-C9, no insensitivity to fungicides other than azoles have been detected in isolates carrying F46Y/M172V/E427K. The remaining clusters, A, B and G, were formed by wild-type CYP51A isolates carrying CSP t04A with MAT1-1, CSP t01 with MAT1-1, and t18A with MAT1-1, respectively, and all showing sensitivity to all fungicides tested.

## 4. Discussion

Tulip bulbs have previously been identified as a vehicle for the international spread of azole resistant *Af* isolates [23,24], and decaying flower bulb waste from farms has already been identified as a hotspot in a previous study [25]. However, little is known about the emergence and spatiotemporal spread of different fungicide resistant alleles belonging to different modes of action, such as MBC, QoI and SDHI fungicides [17], in this environment. In this study, we sampled *Af* populations from tulip field soils, tulip bulbs, tulip peel waste heaps and tulip grower compost and compared their fungicide sensitivity status to both azole and non-azole fungicides. A selection of fungicide insensitive isolates associated with tulip cultivation and a panel of medical isolates were further characterized for presence of fungicide resistant alleles and genotyped using CSP and STR*Af* typing [26,27]. Several clones were identified and the level of clonal expansion of pan-azole and non-azole resistant strains in both the clinical setting and the wider environment is also further discussed hereinafter.

The 5.6% frequency (ten out of 180 isolates) of pan-azole insensitive *Af* strains isolated from Dutch tulip field soils in 2016 (Table 1) was higher than the reported 0.5% (two out 428 isolates) measured for arable soils sampled at nine different locations in France, Germany, and the UK in 2015 using the same sampling and test procedures [17]. Three out of the ten pan-azole insensitive isolates, all sampled from the same tulip field carried TR_46_/Y121F/T289A, while six TR_34_/L98H and one TR_34_/L98H/S297T/F495I strain were identified as well. One TR_34_/L98H, STNL6-B1, and all three TR_46_/Y121F/T289A isolates, STNL5-B7, STNL5-C1 and STNL5-C7, were also insensitive to both the MBC fungicide carbendazim and the QoI fungicide pyraclostrobin (Table 2) which was attributed to alterations in β-tubulin (F200Y) and cytochrome *b* (G143A), respectively, as previously reported [17]. One of the TR_46_/Y121F/T289A strains, STNL5-C8, shared a MLG (26-21-9-33-11-22-8-14-10) with clinical isolates from the Netherlands and environmental isolates from India and Tanzania [35,36]. Interestingly, STNL6-B1 was also insensitive to the SDHI fungicide boscalid and carried the SdhB alteration H270Y, a change associated with SDHI resistance [17]. The TR_34_/L98H/S297T/F495I isolate, STNL5-C5, was also insensitive to carbendazim and showed a double mutation resulting in the F200Y β-tubulin alteration. One F46Y/M172V/E427K isolate, STNL2-C9, showing low levels of insensitivity to azoles, was also moderately insensitive to pyraclostrobin (MIC value of 6.017 µg/mL) and carried cytochrome *b* F129L, an alteration well known to confer low to moderate levels of QoI resistance in plant pathogenic fungi [37], and recently also reported for *Af* [19]. Only one azole sensitive strain, STNL1-A8, carrying wild-type CYP51A appeared to have evolved carbendazim insensitivity by acquiring the beta-tubulin F200Y allele, which has been reported once before [19]. Azole resistance seems to have spread quickly after the emergence of TR_34_- and TR_46_-based CYP51A variants, with subsequent selection within these populations for resistance to fungicides belonging to other modes of actions in environments where multiple classes of fungicides are present. The presence of an azole sensitive isolate with carbendazim insensitivity can be explained by the rare emergence and survival of this strain in an environment with and without MBC fungicides, in the absence of azole fungicides or before azoles were introduced. Alternatively, this isolate could also be the offspring from a cross between a multi-fungicide resistant azole insensitive isolate and a wild-type strain. 

High numbers of *Af* colonies (up to 100 per bulb) were isolated from the tulip bulb samples when sampling the outer peels, which is in accordance with other studies investigating tulip and flower bulb samples [23,24]. The frequency of pan-azole resistant isolates was higher for the population isolated from tulip bulbs with 25 out of 108 isolates (23.1%) detected in 2015. The frequency was higher for bulbs purchased in the Netherlands (36.7%, 22 out of 60) than for bulbs purchased in the UK (6.3%, three out of 48), but the frequency for the UK-produced bulb population could be lower as it is likely that some purchased bulbs were directly imported from the Netherlands and not cultivated in the UK. No pan-azole resistant isolates were detected in populations sampled from Dutch tulip bulbs or UK daffodil bulbs in 2017, although the population size was smaller with 20 and 30 isolates, respectively. Seven out of the 25 pan-azole resistant isolates from the tulip bulb population carried TR_46_/Y121F/T289A and were also highly insensitive to both carbendazim and pyraclostrobin. These seven isolates originated from different tulip bulb samples purchased in the Netherlands and the UK in 2015 and formed together with two Dutch tulip peel waste heap strains, isolated in 2018, and a Dutch environmental reference strain (TR46-NL), isolated before 2014, a cluster (cluster I in Figure 3) with an identical CSP type (t01), mating type (MAT1-1) and MLG (26-21-12-26-9-20-13.3-9-9), indicative for a clonal lineage [28]. The same MLG clone has also been widely reported for both clinical and environmental strains from the Netherlands in the past [35,36], showing its spread and longevity through asexual reproduction. The successful expansion of this clone can be explained by a greater fitness in comparison with other genotypes. The presence of both β-tubulin F200Y and cytochrome *b* G143A alleles, conferring resistance to MBC and QoI fungicides, respectively, will increase the chance of survival in environments where multiple modes of fungicides are used (bulb dipping, seed treatments and foliar sprays) or can be found (plant waste and compost heaps). One pan-azole resistant tulip bulb isolate carrying TR_34_/L98H (T11-8) was also insensitive to carbendazim and pyraclostrobin and can also be considered as a clonal lineage as strains with an identical MLG (14-20-9-31-9-10-8-10-28) have been reported in clinical and environmental strains from India and in environmental strains from Tanzania [36,38]. The successful rapid expansion of this clone in India can be explained by its multi-fungicide resistance status, the ability to outcompete other strains in the presence of MBC and/or QoI fungicides upon its arrival in India. However, the emergence of this strain in India through sexual recombination and subsequent spread to other continents cannot be ruled out [38]. In this context it would be interesting to see if the clonal expansion of a TR_34_/L98H strain (MLG 22-10-9-9-9-23-8-10-8) reported in compost in 13 different cities in Iran can also be explained by the presence of additional fungicide resistant alleles in this clone [39]. The remaining 17 pan-azole resistant isolates were all sensitive to both carbendazim and pyraclostrobin. At least 11 of these carried TR_34_/L98H, from which at least eight were grouped together with a 2016 sugar beet field soil isolate from Belgium in cluster C, having an identical MLG (20-21-16-77-12-11-16.3-11-21), CSP t11 and MAT1-1 in common. As far as we know, this clone has not been reported before. 

As expected, the highest frequency of pan-azole resistant isolates, 52.4%, was detected in samples collected from tulip peel waste heaps with 22 out of 42 isolates detected. The frequency of pan-azole resistant strains in compost samples could not be determined as *Af* isolates were mainly recovered from tebuconazole-amended agar due to excessive growth of other fungi on agar without addition of tebuconazole. High frequencies of up to 21.5% for recovery of pan-azole resistant strains from flower bulb waste heaps has also been reported in earlier studies [25]. Almost all pan-azole resistant strains (21 out of 22) were also highly insensitive to both carbendazim and pyraclostrobin, non-azole fungicides with different modes of action that are commonly found in tulip peel or flower waste heaps because of their application in bulb dipping and/or as part of foliar sprays to control diseases like Fusarium and Botrytis. Further evaluation of tulip peel waste heap and compost isolates showed that most multi-fungicide resistant isolates with high levels of voriconazole insensitivity (MIC value > 19.120 µg/mL) carried TR_46_/Y121F/T289A, but a more complex CYP51A variant, TR_34_/L98H/T289A/I364V/G448S, was also detected in two isolates that were isolated from colonies growing on tebuconazole-amended agar (Table 2). This variant has been found in clinical strains from South Korea [40] and the UK since 2016 [41], but has also been reported in strains isolated from imported Dutch tulip bulbs in Japan in 2018 [24], and from compost heap samples in the UK in 2019 that were sampled as part of a citizen-science project [42]. The emergence of more complex variants over time shows similarities with the evolution of azole resistance in the plant pathogen *Zymoseptoria tritici*, where a stepwise accumulation of *CYP51* mutations has resulted in adaptation to higher azole dose rates and/or new azoles with different binding properties entering the market [43]. 

The collection of clinical isolates used in this study showed a wide variety of fungicide sensitivity pheno- and genotypes that were also found in the environmental isolates. CYP51A variants TR_34_/L98H, TR_34_/L98H/S297T/F495I and TR_46_/Y121F/T289A, were detected in eight, four, eight and six isolates, respectively. The highest levels of insensitivity to voriconazole, imazalil and tebuconazole were generally measured for isolates carrying TR_46_/Y121F/T289A, while TR_34_/L98H/S297T/F495I isolates were generally less sensitive to imazalil than TR_34_/L98H isolates which can be explained by the presence of F495I, which particularly affects the binding of imidazoles such as prochloraz and imazalil [12]. Three out of the four TR_34_/L98H/S297T/F495I isolates carried beta-tubulin F200Y and were highly insensitive to the MBC fungicide carbendazim. All six TR_46_/Y121F/T289A strains carried both beta-tubulin F200Y and cytochrome *b* G143A alleles, the latter conferring high levels of insensitivity to the QoI fungicide pyraclostrobin. One TR_46_/Y121F/T289A strain, CYP_15_38 with a rare CSP type t09, was also insensitive to boscalid, a SDHI fungicide, and carried SdhB H270Y. Two out of eight TR_34_/L98H isolates, ARAF017 and CYP_51_46, were also insensitive to all three non-azole fungicides carbendazim, pyraclostrobin and boscalid. One new fungicide resistant allele was detected, beta-tubulin E198A in isolate CYP_51_46, conferring resistance to carbendazim. This allele, like beta-tubulin F200Y, commonly found in plant pathogens that developed resistance to MBC fungicides [44], has recently been reported in *Af* [17]. 

STR*Af* typing of the clinical isolates revealed three additional clonal lineages. The German clinical isolate Asp 261, carrying TR_34_/L98H/S297T/F495I and beta-tubulin F200Y, shared an identical MLG (14-10-9-30-9-6-8-10-20) with clinical strain Afu_key29 reported in Denmark [12]. Two clinical TR_46_/Y121F/T289A isolates, V093-54 and CYP_15_7 from the Netherlands and Belgium, respectively, shared the same MLG (26-21-9-33-11-22-8-14-10), CSP type (t01) and mating type (MAT1-2) with the Dutch tulip field soil isolate STNL5-C8 and all three isolates carried beta-tubulin F200Y and cytochrome *b* G143A alleles. Finally, tulip peel waste heap isolate TP UT5C-5 and clinical isolate OKH50, isolated in Japan in 2016, shared the same MLG (14-21-8-31-9-6-8-10-20) (cluster F in Figure 3) with an Irish clinical strain D5 isolated in 2015 [23], all carrying TR_34_/L98H. Traits like CSP t02 and MAT1-2 were shared between TP UT5C-5 and OKH50, but only TP UT5C-5 was highly insensitive to pyraclostrobin. This shows that pan-azole resistant clones of *Af* can spread and develop de novo resistant alleles under exposure to different fungicides in hotspots. Using MLGs only based on STR*Af* typing might underestimate the spread of clones as low levels of instability have recently been reported for markers 3A and 3C. The recently reported MLG (14-21-8-32-9-6-8-10-20) for Chinese farm soil isolates carrying TR_34_/L98H could also represent descendants of the same clone as OKH50 [45]. Whole genome sequencing and/or further typing using additional markers targeting hypervariable TRs within exons of surface protein coding genes will be needed to confirm this [41,46]. 

## 5. Conclusions

The understanding of the emergence and epidemics of pan-azole resistant *Af* over time has been improved by this and other recently published research linking strains found in patients with genotypes distributed in the wider environment [41,47]. The acquisition of azole-resistant alleles in clinical and/or environmental settings, as well as non-azole fungicide resistant alleles in the environment, have contributed to the rapid expansion of clonal lineages in hotspots under selection by fungicides, where sexual reproduction can also generate new genotypes with combinations of fungicide resistant alleles. Continuous real-time surveillance of aerosol samples at different locations for the presence of *Af* combined with identification of CYP51A variants and other fungicide resistant alleles using a NGS genomics approach can provide information on the emergence and spatiotemporal dynamics of newly evolved strains. The detection of newly evolved azole resistant strains and clones can also be used to identify hotspots, where measures can be devised to reduce reproduction and/or fungicide selection pressure, aiming to minimize the risk of human exposure to airborne pan-azole resistant *Af* conidia through inhalation. Strains carrying novel CYP51A variants, based on new mutations and or combinations of (new) mutations, as well as strains harboring different azole resistance mechanisms [48], should be tested against all available azole antifungals, as changes in their azole susceptibility status can guide decisions for an optimal diagnosis and management of aspergillosis in patients [49].

## Figures and Tables

**Figure 1 microorganisms-09-02379-f001:**
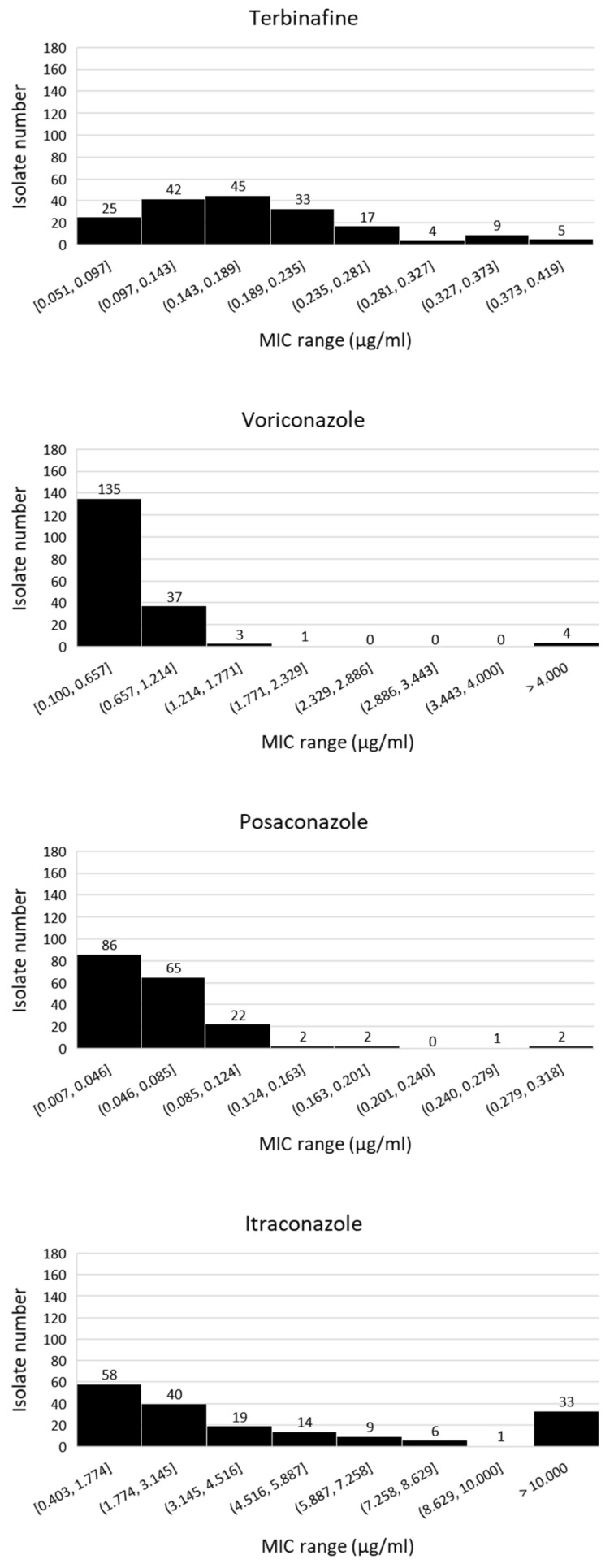
MIC distribution of *A. fumigatus* isolates sampled from tulip field soils (*n* = 180) in the Netherlands. Numbers show amount (sum) of isolates within each MIC range.

**Figure 2 microorganisms-09-02379-f002:**
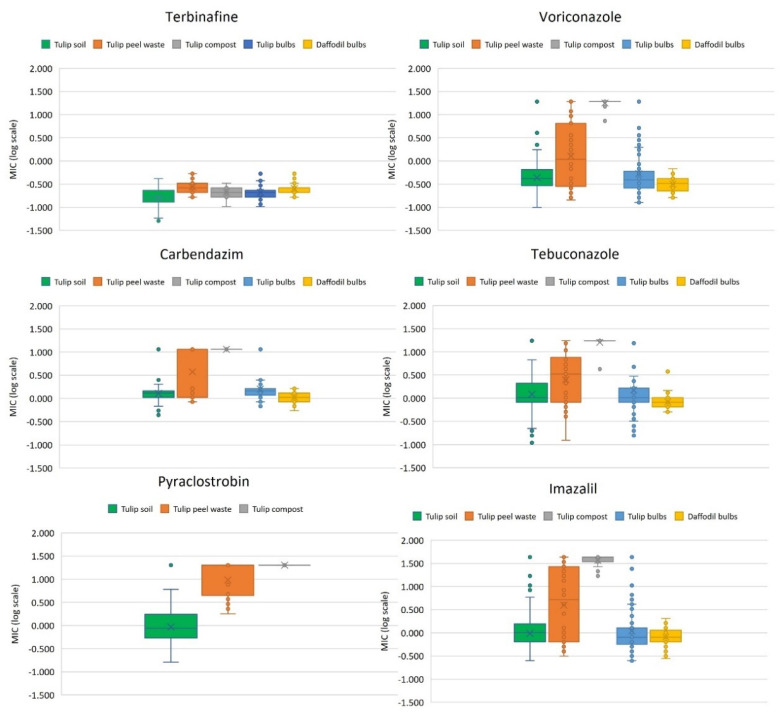
Comparison of fungicide sensitivity levels for strains isolated from tulip field soils (*n* = 180), tulip peel waste (*n* = 42), tulip compost (*n* = 19) and flower bulbs (tulips (*n* = 128) and daffodils (*n* = 30)). For voriconazole, tebuconazole and carbendazim out of range values spiral plating MIC values were displayed as the log values of 19.120, 17.349 and 11.464 µg/mL, respectively.

**Figure 3 microorganisms-09-02379-f003:**
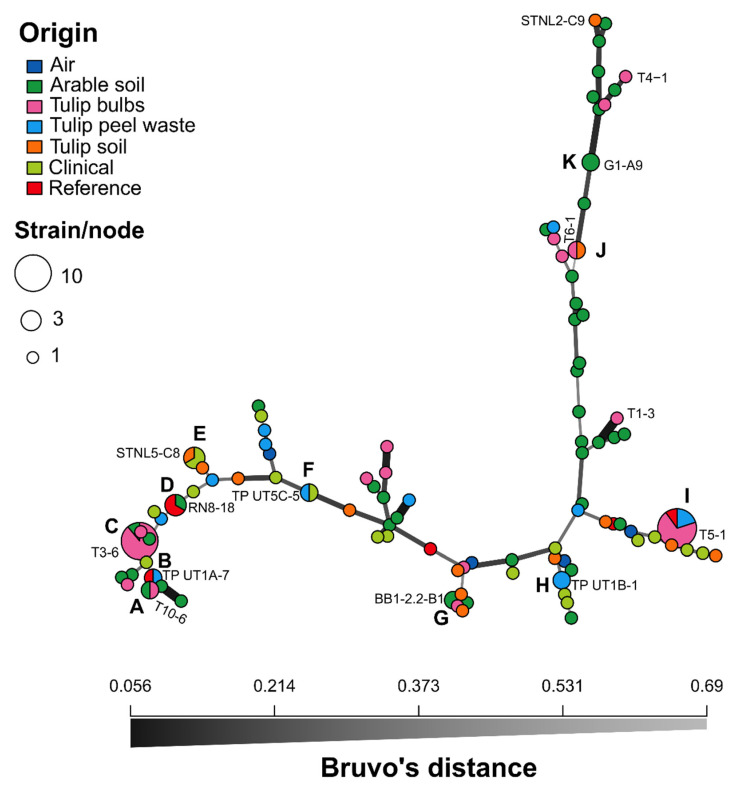
Minimum-spanning networks showing the genetic relationship of *Aspergillus fumigatus* multilocus genotypes (MLGs) originating from different sources. The relatedness between MLGs is based on Bruvo’s genetic distances, which accounts for the stepwise mutation of microsatellite loci. Each node represents an MLG with one or more individuals. Nodes with multiple isolates (clonal lineages), clusters, are arranged alphabetically. Nodes that are more closely related have darker and thicker edges, whereas nodes that are more distantly related have lighter and thinner edges. Names of a selection of isolates, including all medical strains, are also displayed and all details on strains can be found in Appendix A.

**Table 1 microorganisms-09-02379-t001:** Collected samples with their locations.

Sample ^1^	Origin and Year	Frequency of Azole Resistant Isolates ^2^
Soil tulip field 1	Biddinghuizen, NL (2016)	0/30
Soil tulip field 2	Dronten, NL (2016)	1/30
Soil tulip field 3	Venhuizen, NL (2016)	1/30
Soil tulip field 4	Westwoud, NL (2016)	0/30
Soil tulip field 5	Westwoud, NL (2016)	5/30
Soil tulip field 6	Zyperdijk, NL (2016)	3/30
Bulbs 1 Tulipa ‘Stresa’	Noordwijkerhout, NL (2015)	4/10
Bulbs 2 Tulipa ‘Apeldoorn Yellow’	Vaassen, NL (2015)	4/10
Bulbs 3 Tulipa greigii ‘Roodkapje’	Vaassen, NL (2015)	4/10
Bulbs 4 Tulipa ‘Praestans Shogun’	Vaassen, NL (2015)	3/10
Bulbs 5 Tulipa ‘Claudia’	Noordwijkerhout, NL (2015)	6/10
Bulbs 6 Tulipa ‘Triumph Hotpants’	Millbrook, UK (2015)	1/8
Bulbs 7 Tulipa ‘Mickey Mouse’	Canterbury, UK (2015)	1/10
Bulbs 8 Tulipa ‘Gavota’	Wickford, UK (2015)	0/10
Bulbs 9 Tulipa ‘Guiseppi Verdi’	Horsham, UK (2015)	0/10
Bulbs 10 Tulipa ‘White Marvel’	Preston, UK (2015)	1/10
Bulbs 11 Tulipa ‘Red Impression’	Hillegom, NL (2015)	1/10
Bulbs 12 Tulipa ‘Negrita’	Vaassen, NL (2017)	0/10
Bulbs 13 Tulipa ‘Rembrand’	Vaassen, NL (2017)	0/10
Bulbs 14 Narcissus ‘Pink Pride’	Vaassen, NL (2017)	0/10
Bulbs 15 Narcissus ‘Jetfire’	Vaassen, NL (2017)	0/10
Bulbs 16 Narcissus mix	Hillegom, NL (2017)	0/10
Bulb peel waste heap tulip grower 1	NL (2018)	6/11
Bulb peel waste heap tulip grower 2	NL (2018)	1/1
Bulb peel waste heap tulip grower 3	NL (2018)	1/2
Bulb peel waste heap tulip grower 4	NL (2018)	3/4
Bulb peel waste heap tulip grower 5	NL (2018)	11/19
Bulb peel waste heap tulip grower 6	NL (2018)	1/5
Compost heap tulip grower A	NL (2018)	8/8
Compost heap tulip grower B	NL (2018)	1/1
Compost heap tulip grower C	NL (2018)	10/10

^1^ Isolates were obtained from colonies growing on Sabouraud Dextrose agar plates without fungicides with exception of the compost heap isolates that were picked from plates amended with 5 µg/mL tebuconazole. ^2^ Azole resistant isolates have at least two raised MIC values for either voriconazole (>1.0 µg/mL), imazalil (>2.0 µg/mL) and/or tebuconazole (>4.0 µg/mL).

**Table 2 microorganisms-09-02379-t002:** Sensitivity of environmental *Aspergillus fumigatus* isolates to a panel of fungicides belonging to different modes of action and their further characterization using CSP, mating type and CYP51A variant analysis. Isolates ranked according to voriconazole sensitivity (low to high MIC values in µg/mL).

Isolate	CYP51A	VRC	IMA	TEB	CAR	PYR	BOS	CSP	Mating Type
STNL1-A8	WT	0.292	0.714	0.814	>11.464	1.110	-	t04A	MAT1-1
T8-2	WT	0.471	0.803	0.643	1.169	0.872	0.253	t11	MAT1-1
STNL5-C5	TR_34_/L98H/S297T/F495I	0.531	8.367	4.227	>11.464	0.685	0.199	t02	MAT1-2
T6-3	TR_34_/L98H	0.675	5.236	>17.349	2.013	0.477	-	t11	MAT1-1
STNL2-C9	F46Y/M172V/E427K	0.760	1.443	2.970	0.843	6.017	-	t02B	MAT1-1
STNL3-C8	TR_34_/L98H	0.857	1.283	4.227	1.303	>20.120	-	t04B	MAT1-1
TP UT1A-2	TR_34_/L98H	0.857	8.367	5.348	>11.464	>20.120	0.253	t02	MAT1-2
STNL6-B2	TR_34_/L98H	1.381	2.592	4.227	1.169	0.538	0.157	-	MAT1-1
STNL6-A3	TR_34_/L98H	1.381	3.277	6.768	1.303	0.538	0.199	t01	MAT1-2
T3-6	TR_34_/L98H	1.381	6.619	>17.349	2.013	1.110	0.253	t11	MAT1-1
TP UT1B-1	TR_34_/L98H	1.754	6.619	>17.349	>11.464	>20.120	0.517	t11	MAT1-1
TP UT5C-5	TR_34_/L98H	1.754	6.619	4.227	0.679	>20.12	0.224	t02	MAT1-2
STNL2-B8	TR_34_/L98H	1.754	10.577	4.227	2.013	0.294	0.321	t04B	MAT1-1
STNL5-B6	TR_34_/L98H	2.227	1.622	3.341	1.619	0.872	0.321	t11	MAT1-2
T2-1	TR_34_/L98H	2.227	4.142	>17.349	1.619	1.110	0.321	t11	MAT1-1
STNL6-B1	TR_34_/L98H	4.047	5.887	>17.349	>11.464	>20.120	>18.469	t01	MAT1-2
T11-8	TR_34_/L98H	5.139	10.577	>17.349	>11.464	>20.120	0.517	t02	MAT1-1
STNL5-B7	TR_46_/Y121F/T289A	>19.120	16.901	>17.349	>11.464	>20.120	0.157	t02	MAT1-1
T7-9	TR_46_/Y121F/T289A	>19.120	24.020	>17.349	>11.464	>20.120	-	t01	MAT1-1
TP UT4A-1	TR_46_/Y121F/T289A	>19.120	34.138	>17.349	>11.464	>20.120	0.407	t01	-
STNL5-C1	TR_46_/Y121F/T289A	>19.120	>43.153	>17.349	>11.464	>20.120	0.177	t06A	MAT1-1
STNL5-C8	TR_46_/Y121F/T289A	>19.120	>43.153	>17.349	>11.464	>20.120	0.199	-	MAT1-2
T3-5	TR_46_/Y121F/T289A	>19.120	>43.153	>17.349	>11.464	>20.120	-	t01	MAT1-1
T4-9	TR_46_/Y121F/T289A	>19.120	>43.153	>17.349	>11.464	>20.120	-	t01	MAT1-1
T4-10	TR_46_/Y121F/T289A	>19.120	>43.153	>17.349	>11.464	>20.120	-	t01	MAT1-1
T5-1	TR_46_/Y121F/T289A	>19.120	>43.153	>17.349	>11.464	>20.120	0.285	t01	MAT1-1
T5-2	TR_46_/Y121F/T289A	>19.120	>43.153	>17.349	>11.464	>20.120	-	t01	MAT1-1
T5-5	TR_46_/Y121F/T289A	>19.120	>43.153	>17.349	>11.464	>20.120	-	t01	MAT1-1
TP TEB5C-2	TR_34_/L98H/T289A/I364V/G448S	>19.120	>43.153	>17.349	>11.464	>20.120	0.321	t02	MAT1-2
TC TEB6B-1	TR_34_/L98H/T289A/I364V/G448S	>19.120	>43.153	>17.349	>11.464	>20.120	>18.469	t02	MAT1-2

VRC (voriconazole), IMA (imazalil) and TEB (tebuconazole) are azoles, inhibiting 14α-demethylase (sterol biosynthesis); CAR (carbendazim) is a MBC fungicide, inhibiting β-tubulin assembly (cytoskeleton); PYR (pyraclostrobin) is a QoI fungicide, inhibiting respiration (complex III); BOS (boscalid) is a SDHI fungicide, inhibiting respiration (complex II); -, not determined.

**Table 3 microorganisms-09-02379-t003:** Sensitivity of clinical *Aspergillus fumigatus* isolates to a panel of fungicides belonging to different modes of action and their further characterization using CSP, mating type and CYP51A variant analysis. Isolates ranked according to voriconazole sensitivity (low to high MIC values in µg/mL).

Isolate	CYP51A	VOR	IMA	TEB	CAR	PYR	BOS	CSP	Mating Type
AF65	WT	0.329	1.141	0.814	1.619	1.413	0.157	t02	MAT1-2
AF293	F46Y/M172V/N284T/D255E/E427K	0.531	2.915	1.649	1.303	0.538	-	t06A	MAT1-2
ARAF013	TR_34_/L98H/S297T/F495I	0.760	13.370	4.227	>11.464	0.332	0.253	t11	MAT1-2
Asp 251	TR_34_/L98H/S297T/F495I	1.088	19.002	5.348	>11.464	1.413	0.157	t02	MAT1-1
CYP_15_63	TR_34_/L98H/S297T/F495I	1.088	13.370	6.768	>11.464	1.110	1.702	t01	MAT1-1
D007	TR_34_/L98H/S297T/F495I	1.088	15.032	3.341	0.843	0.100	0.285	t04A	MAT1-1
CXH_07	TR_34_/L98H	1.381	5.887	6.768	1.303	0.332	0.199	t04A	MAT1-1
Asp 267	TR_34_/L98H	1.381	4.142	4.227	1.619	0.161	0.157	t11	MAT1-2
CXH_06	TR_34_/L98H	1.557	5.236	8.563	1.303	0.423	0.321	t04B	MAT1-2
Asp 164	TR_34_/L98H	1.976	6.619	6.768	1.619	1.110	0.224	t11	MAT1-2
Asp 168	TR_34_/L98H	1.976	5.236	8.563	2.013	0.607	0.199	t04B	MAT1-2
OKH50	TR_34_/L98H	2.227	4.142	5.348	0.757	0.332	0.407	t02	MAT1-2
ARAF017	TR_34_/L98H	4.560	13.370	>17.349	>11.464	>20.12	>18.469	t04A	MAT1-2
CYP_15_46	TR_34_/L98H	5.791	8.367	13.711	>11.464	>20.12	>18.469	t02	MAT1-1
CYP_15_80	TR_46_/Y121F/T289A	9.338	>43.153	8.563	>11.464	>20.12	0.199	t02	MAT1-1
CYP_15_2	TR_46_/Y121F/T289A	>19.12	>43.153	13.711	>11.464	>20.12	0.253	t01	MAT1-2
CYP_15_7	TR_46_/Y121F/T289A	>19.12	>43.153	>17.349	>11.464	>20.12	0.253	t01	MAT1-2
CYP_15_38	TR_46_/Y121F/T289A	>19.12	>43.153	>17.349	>11.464	>20.12	>18.469	t09	MAT1-2
V093-26	TR_46_/Y121F/T289A	>19.12	27.006	8.563	>11.464	>20.12	0.517	t01	MAT1-2
V094-54	TR_46_/Y121F/T289A	>19.12	>43.153	>17.349	>11.464	>20.12	0.285	t01	MAT1-2

VRC (voriconazole), IMA (imazalil) and TEB (tebuconazole) are azoles, inhibiting 14α-demethylase (sterol biosynthesis); CAR (carbendazim) is a MBC fungicide, inhibiting β-tubulin assembly (cytoskeleton); PYR (pyraclostrobin) is a QoI fungicide, inhibiting respiration (complex III); BOS (boscalid) is a SDHI fungicide, inhibiting respiration (complex II); -, not determined.

## Data Availability

Data are available upon request.

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
