# Peer review of "Epidemiological Studies of Pan-Azole Resistant Aspergillus fumigatus Populations Sampled during Tulip Cultivation Show Clonal Expansion with Acquisition of Multi-Fungicide Resistance as Potential Driver"

_microorganisms, 2021, doi:10.3390/microorganisms9112379_

Round 1

Reviewer 1 Report

this study addresses resistance development to clinically relevant azoles  in Af isolates from different sources and of additional mutations to antifungals used in agriculture specifically. Clonal expansion can be observed and seems correlated with increased resistance to other antifungals used in agriculture specifically. The data are relevant in understanding how clonal expansion could occur in nature. 

the author should address the following points

line 102-105: ppm is used as unit instead of MIC in ug/mL that is used regularly in medical setting, this makes interpretation to medical settings rather difficult : values should be converted to concentration in ug/mL and the breakpoints reported in medical setting should  be described or referred to as well in order to understand the relationship. Furtermore: raised two above MIC values: authors should indicate if these represent MIC50 or MIC90 etc

line 112 and 115: is this the same buffer as indicated above, use abbreviation or  describe composition buffer

line 119 refer to references when clinical isolates were previously described 

line 185-186 rephrase line, it does not read well: , three (...), all.... : I believe the authors mean three (...) isolates from ....

line 201-202 what is tested here ? explain and/or refer to figure/table. why is posaconazole most effective?

line 207 and at many other places: write A fumigatus in italics

line 213: why are these not in table 1?

ine 219 were are these numbers (18,17 and 14) shown or described, fig 2 is not providing this info? the same for the numbers described below; refer to figure or table
throughout this result part it is very hard to follow which isolates have specific mutations and the amount of isolates with specific resistance from different sources; this must be optimized in the text

line 310 : RF is not explained

the discussion summarizes many isolates with specific names and mutations, this is particularly difficult to read. authors should optimize the presentation in the discussion. Use an additional figure or table to visualize data and results mote strongly

Author Response

Thanks for the feedback and useful comments. Please see the attachment for the response.

Reviewer 2 Report

The topic is interesting and the manuscript is well prepared. Only a few comments.

The first section of the discussion should summarize the main results obtained in the study and then compare it with previous literature. Please, revise.

the conclusions should stress a bit more the impact of these results in terms of public health and humans' health

Author Response

Thanks for the feedback and useful comments. Please see attachment for the response.
